# LAMP: Extracting Text from Gradients with Language Model Priors

**Mislav Balunović**,* **Dimitar I. Dimitrov**,* **Nikola Jovanović, Martin Vechev**
{mislav.balunovic,dimitar.iliev.dimitrov,
nikola.jovanovic,martin.vechev}@inf.ethz.ch
Department of Computer Science
ETH Zurich

## Abstract

Recent work shows that sensitive user data can be reconstructed from gradient updates, breaking the key privacy promise of federated learning. While success was demonstrated primarily on image data, these methods do not directly transfer to other domains such as text. In this work, we propose LAMP, a novel attack tailored to textual data, that successfully reconstructs original text from gradients. Our attack is based on two key insights: (i) modeling prior text probability with an auxiliary language model, guiding the search towards more natural text, and (ii) alternating continuous and discrete optimization, which minimizes reconstruction loss on embeddings, while avoiding local minima by applying discrete text transformations. Our experiments demonstrate that LAMP is significantly more effective than prior work: it reconstructs 5x more bigrams and $23\%$ longer subsequences on average. Moreover, we are the first to recover inputs from batch sizes larger than 1 for textual models. These findings indicate that gradient updates of models operating on textual data leak more information than previously thought.

## 1 Introduction

Federated learning [24] (FL) is a widely adopted framework for training machine learning models in a decentralized way. Conceptually, FL aims to enable training of highly accurate models without compromising client data privacy, as the raw data never leaves client machines. However, recent work [28, 43, 41] has shown that the server can in fact recover the client data, by applying a reconstruction attack on the gradient updates sent from the client during training. Such attacks typically start from a randomly sampled input and modify it such that its corresponding gradients match the gradient update originally sent by the client. While most works focus on reconstruction attacks in computer vision, there has comparatively been little work in the text domain, despite the fact that some of the most prominent applications of FL involve learning over textual data, e.g., next-word prediction on mobile phones [30]. A key component of successful attacks in vision has been the use of image priors such as total variation [7]. These priors guide the reconstruction towards natural images, which are more likely to correspond to client data. However, the use of priors has so far been missing from attacks on text [43, 3], limiting their ability to reconstruct real client data.

**This work: private text reconstruction with priors** In this work, we propose LAMP, a new reconstruction attack which leverages language model priors to extract private text from gradients. The overview of our attack is given in Fig. 1. The attacker has access to a snapshot of the transformer network being trained in a federated manner (e.g., BERT), and a gradient $\nabla_\theta \mathcal{L}(\boldsymbol{x}^*, y^*)$ which the client has computed on that snapshot, using their private data. The attack starts by sampling token

---

*Equal contribution.

36th Conference on Neural Information Processing Systems (NeurIPS 2022).

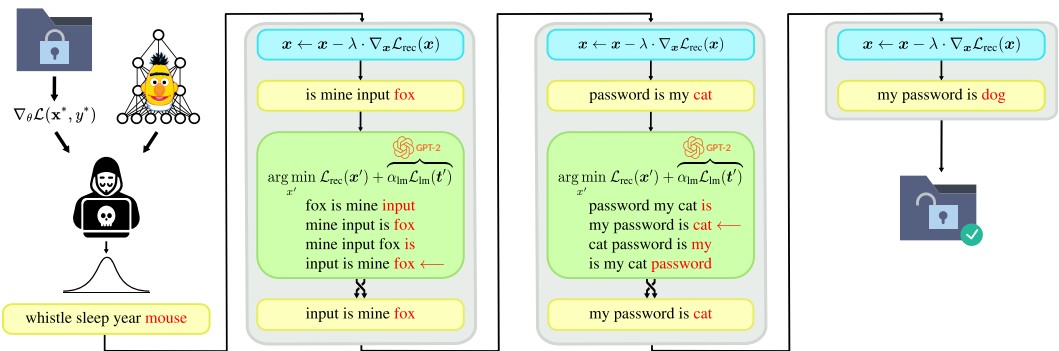

Figure 1: An overview of LAMP. We initialize the reconstruction by sampling from a Gaussian distribution, and alternate between continuous and discrete optimization. Continuous optimization minimizes the reconstruction loss with an embedding regularization term. Discrete optimization forms candidates by applying transformations, and chooses the best candidate based on a combination of reconstruction loss and perplexity, as measured by an auxiliary language model (e.g., GPT-2).

embeddings from a Gaussian distribution to create the initial reconstruction. Then, at each step, we improve the reconstruction (shown in yellow) by alternating between continuous (blue) and discrete optimization (green). The continuous part minimizes the reconstruction loss, which measures how close the gradients of the current reconstruction are to the observed client gradients, together with an embedding regularization term. However, this is insufficient as the gradient descent can get stuck in a local optimum due to its inability to make discrete changes to the reconstruction. We address this issue by introducing a discrete step—namely, we generate a list of candidate sentences using several transformations on the sequence of tokens (e.g., moving a token) and select a candidate that minimizes the combined reconstruction loss and perplexity, which measures the likelihood of observing the text in a natural distribution. We use GPT-2 [29] as an auxiliary language model to measure the perplexity of each candidate (however, our method allows using other models). Our final reconstruction is computed by setting each embedding to its nearest neighbor from the vocabulary.

Key component of our reconstruction attack is the use of a language model prior combined with a search that alternates continuous and discrete optimization steps. Our experimental evaluation demonstrates the effectiveness of this approach—LAMP is able to extract text from state-of-the-art transformer models on several common datasets, reconstructing up to 5 times more bigrams than prior work. Moreover, we are the first to perform text reconstruction in more complex settings such as batch sizes larger than 1, fine-tuned models, and defended models. Overall, across all settings we demonstrate that LAMP is effective in reconstructing large portions of private text.

**Main contributions**   Our main contributions are:

- LAMP, a novel attack for recovering input text from gradients, which leverages an auxiliary language model to guide the search towards natural text, and a search procedure which alternates continuous and discrete optimization.

- An implementation of LAMP and its extensive experimental evaluation, demonstrating that it can reconstruct significantly more private text than prior work. We make our code publicly available at `https://github.com/eth-sri/lamp`.

- The first thorough experimental evaluation of text attacks in more complex settings such as larger batch sizes, fine-tuned models and defended models.

## 2   Related Work

Federated learning [24, 18] has attracted substantial interest [16] due to its ability to train deep learning models in a decentralized way, such that individual user data is not shared during training. Instead, individual clients calculate local gradient updates on their private data, and share them with a centralized server, which aggregates them to update the model [24]. The underlying assumption is that user data cannot be recovered from gradient updates. Recently, several works [28, 43, 41, 42]

demonstrated that gradients can in fact still leak information, invalidating the fundamental privacy assumption. Moreover, recent work achieved near-perfect image reconstruction from gradients [7, 40, 14]. Interestingly, prior work showed that an auxiliary model [14] or prior information [2] can significantly improve reconstruction quality. Finally, Huang et al. [12] noticed that gradient leakage attacks often make strong assumptions, namely that batch normalization statistics and ground truth labels are known. In our work, we do not assume knowledge of batch normalization statistics and as we focus on binary classification tasks, we can simply enumerate all possible labels.

Despite substantial progress on image reconstruction, attacks in other domains remain challenging, as the techniques used for images rely extensively on domain specific knowledge. In the domain of text, in particular, where federated learning is often applied [32], only a handful of works exist [43, 3, 22]. DLG [43] was first to attempt reconstruction from gradients coming from a transformer; TAG [3] extended DLG by adding an $L_1$ term to the reconstruction loss; finally, unlike TAG and DLG which are optimization-based techniques, APRIL [22] recently demonstrated an exact gradient leakage technique applicable to transformer networks. However, APRIL assumes batch size of $1$ and learnable positional embeddings, which makes it simple to defend against. Another attack on NLP is given in Fowl et al. [6], but they use stronger assumption that the server can send malicious updates to the clients. Furthermore, there is a concurrent work [11] on reconstructing text from transformers, but it is limited to the case when token embeddings are trained together with the network.

Finally, there have been several works attempting to protect against gradient leakage. Works based on heuristics [34, 31] lack privacy guarantees and have been shown ineffective against stronger attacks [2], while those based on differential privacy do train models with formal guarantees [1], but typically hurt the accuracy of the trained models as they require adding noise to the gradients. We remark that we also evaluate LAMP on defended networks.

# 3 Background

In this section, we introduce the background necessary to understand our work.

## 3.1 Federated Learning

In federated learning, $C$ clients aim to jointly optimize a neural network $f$ with parameters $\boldsymbol{\theta}$ on their private data. At iteration $k$, the parameters $\boldsymbol{\theta}^k$ are sent to all clients, where each client $c$ executes a gradient update $\nabla_{\boldsymbol{\theta}^k} \mathcal{L}(\boldsymbol{x}_c^*, y_c^*)$ on a sample $(\boldsymbol{x}_c^*, y_c^*)$ from their dataset $(\mathcal{X}_c, \mathcal{Y}_c)$. The updates are sent back to the server and aggregated. While in Sec. 5 we experiment with both FedSGD [24] and FedAvg [19] client updates, throughout the text we assume that clients use FedSGD updates:

$$\boldsymbol{\theta}^{k+1} = \boldsymbol{\theta}^k - \frac{\lambda}{C} \sum_{c=1}^{C} \nabla_{\boldsymbol{\theta}^k} \mathcal{L}(\boldsymbol{x}_c^*, y_c^*).$$

**Gradient leakage attacks** A gradient leakage attack is an attack executed by the server (or a party which compromised it) that tries to obtain the private data $(\boldsymbol{x}_c^*, y_c^*)$ of a client using the gradient updates $\nabla_{\boldsymbol{\theta}^k} \mathcal{L}(\boldsymbol{x}_c^*, y_c^*)$ sent to the server. Gradient leakage attacks usually assume honest-but-curious servers which are not allowed to modify the federated training protocol outlined above. A common approach, adopted by Zhu et al. [43], Zhao et al. [41], Deng et al. [3] as well as our work, is to obtain the private data by solving the optimization problem:

$$\underset{(\boldsymbol{x}_c, y_c)}{\arg \min} \ \delta \left( \nabla_{\boldsymbol{\theta}^k} \mathcal{L} \left( \boldsymbol{x}_c^*, y_c^* \right), \nabla_{\boldsymbol{\theta}^k} \mathcal{L} \left( \boldsymbol{x}_c, y_c \right) \right),$$

where $\delta$ is some distance measure and $(\boldsymbol{x}_c, y_c)$ denotes dummy data optimized using gradient descent to have similar gradients $\nabla_{\boldsymbol{\theta}^k} \mathcal{L}(\boldsymbol{x}_c, y_c)$ to true data $(\boldsymbol{x}_c^*, y_c^*)$. Common choices for $\delta$ are $L_2$ [43], $L_1$ [3] and cosine distances [7]. When the true label $y_c^*$ is known, the problem reduces to

$$\underset{\boldsymbol{x}_c}{\arg \min} \ \delta \left( \nabla_{\boldsymbol{\theta}^k} \mathcal{L} \left( \boldsymbol{x}_c^*, y_c^* \right), \nabla_{\boldsymbol{\theta}^k} \mathcal{L} \left( \boldsymbol{x}_c, y_c^* \right) \right),$$

which was shown [41, 12] to be simpler to solve with gradient descent approaches.

## 3.2 Transformer Networks

In this paper, we focus on the problem of gradient leakage of text on transformers [35]. Given some input text, the first step is to tokenize it into tokens from some fixed vocabulary of size $V$. Each token is then converted to a 1-hot vector denoted $\boldsymbol{t}_1, \boldsymbol{t}_2, \ldots, \boldsymbol{t}_n \in \mathbb{R}^V$, where $n$ is the number of tokens in the text. The tokens are then converted to embedding vectors $\boldsymbol{x}_1, \boldsymbol{x}_2, \ldots, \boldsymbol{x}_n \in \mathbb{R}^d$, where $d$ is a chosen embedding size, by multiplying by a trained embedding matrix $\boldsymbol{W}_{\text{embed}} \in \mathbb{R}^{V \times d}$ [4]. The rows of $\boldsymbol{W}_{\text{embed}}$ represent the embeddings of the tokens, and we denote them as $\boldsymbol{e}_1, \boldsymbol{e}_2, \ldots, \boldsymbol{e}_V \in \mathbb{R}^d$. In addition to tokens, their positions in the sequence are also encoded using the positional embedding matrix $\boldsymbol{W}_{\text{pos}} \in \mathbb{R}^{P \times d}$, where $P$ is the longest allowed token sequence. The resulting positional embeddings are denoted $\boldsymbol{p}_1, \boldsymbol{p}_2, \ldots, \boldsymbol{p}_n$. For notational simplicity, we denote $\boldsymbol{e} = \boldsymbol{e}_1, \boldsymbol{e}_2, \ldots, \boldsymbol{e}_V$, $\boldsymbol{x} = \boldsymbol{x}_1, \boldsymbol{x}_2, \ldots, \boldsymbol{x}_n$ and $\boldsymbol{p} = \boldsymbol{p}_1, \boldsymbol{p}_2, \ldots, \boldsymbol{p}_n$. We use the token-wise sum of the embeddings $\boldsymbol{x}$ and $\boldsymbol{p}$ as an input to a sequence of self-attention layers [35]. The final classification output is given by the first output of the last self-attention layer that undergoes a final linear layer, followed by a $\tanh$.

## 3.3 Calculating Perplexity on Pretrained Models

In this work, we rely on large pretrained language models, such as GPT-2 [29], to assess the quality of the text produced by the continuous part of our optimization. Such models are typically trained to calculate the probability $P(\boldsymbol{t}_n \mid \boldsymbol{t}_1, \boldsymbol{t}_2, ..., \boldsymbol{t}_{n-1})$ of inserting a token $\boldsymbol{t}_n$ from the vocabulary of tokens to the end of the sequence of tokens $\boldsymbol{t}_1, \boldsymbol{t}_2, ..., \boldsymbol{t}_{n-1}$. Therefore, such models can be leveraged to calculate the likelihood of a sequence of tokens $\boldsymbol{t}_1, \boldsymbol{t}_2, ..., \boldsymbol{t}_n$, as follows:

$$P(\boldsymbol{t}_1, \boldsymbol{t}_2, ..., \boldsymbol{t}_n) = \prod_{l=0}^{n-1} P(\boldsymbol{t}_{l+1} \mid \boldsymbol{t}_1, \boldsymbol{t}_2, ..., \boldsymbol{t}_l).$$

One can use the likelihood, or the closely-related negative log-likelihood, as a measure of the quality of a produced sequence. However, the likelihood depends on the length of the sequence, as probability decreases with length. To this end, we use the perplexity measure [13], defined as:

$$\mathcal{L}_{\text{lm}}(\boldsymbol{t}_1, \boldsymbol{t}_2, ..., \boldsymbol{t}_n) = -\frac{1}{n} \sum_{l=0}^{n-1} \log P(\boldsymbol{t}_{l+1} \mid \boldsymbol{t}_1, \boldsymbol{t}_2, ..., \boldsymbol{t}_l).$$

In the discrete part of our optimization, we rely on this measure to assess the quality of reconstructed sequences produced by the continuous part.

# 4 Extracting Text with LAMP

In this section we describe the details of our attack which alternates between continuous optimization using gradient descent, presented in Sec. 4.2, and discrete optimization using language models to guide the search towards more natural text reconstruction, presented in Sec. 4.3.

## 4.1 Notation

We denote the attacked neural network and its parameters with $f$ and $\theta$, respectively. Further, we denote the client token sequence and its label as $(\boldsymbol{t}^*, y^*)$, and our reconstructions as $(\boldsymbol{t}, y)$. For each token $\boldsymbol{t}_i^*$ in $\boldsymbol{t}^*$ and $\boldsymbol{t}_i$ in $\boldsymbol{t}$, we denote their embeddings with $\boldsymbol{x}_i^* \in \mathbb{R}^d$ and $\boldsymbol{x}_i \in \mathbb{R}^d$, respectively. Moreover, for each token in our vocabulary, we denote the embedding with $\boldsymbol{e}_i \in \mathbb{R}^d$. We collect the individual embeddings $\boldsymbol{x}_i, \boldsymbol{x}_i^*$, and $\boldsymbol{e}_i$ into the matrices $\boldsymbol{x} \in \mathbb{R}^{d \times n}$, $\boldsymbol{x}^* \in \mathbb{R}^{d \times n}$ and $\boldsymbol{e} \in \mathbb{R}^{d \times V}$, where $n$ is the number of tokens in $\boldsymbol{t}^*$ and $V$ is the size of the vocabulary.

## 4.2 Continuous Optimization

We now describe the continuous part of our attack (blue in Fig. 1). Throughout the paper, we assume knowledge of the ground truth label $y^*$ of the client token sequence we aim to reconstruct, meaning $y = y^*$. This assumption is not a significant restriction as we mainly focus on binary classification, with batch sizes such that trying all possible combinations of labels is feasible. Moreover, prior work [9, 40] has demonstrated that labels can easily be recovered for basic network architectures, which can be adapted for transformers in future work. We initialize our reconstruction candidate by sampling embeddings from a Gaussian and pick the one with the smallest reconstruction loss.

**Reconstruction loss**    A key component of our attack is a loss measuring how close the reconstructed gradient is to the true gradient. Assuming an $l$-layer network, where $\boldsymbol{\theta}_i$ denotes the parameters of layer $i$, an option is to use the combination of $L_2$ and $L_1$ loss proposed by Deng et al. [3],

$$\mathcal{L}_{\text{tag}}(\boldsymbol{x}) = \sum_{i=1}^{l} ||\nabla_{\boldsymbol{\theta}_i} f(\boldsymbol{x}^*, y^*) - \nabla_{\boldsymbol{\theta}_i} f(\boldsymbol{x}, y)||_2 + \alpha_{\text{tag}} ||\nabla_{\boldsymbol{\theta}_i} f(\boldsymbol{x}^*, y^*) - \nabla_{\boldsymbol{\theta}_i} f(\boldsymbol{x}, y)||_1.$$

where $\alpha_{\text{tag}}$ is a hyperparameter. Another option is to use the cosine reconstruction loss proposed by Geiping et al. [7] in the image domain:

$$\mathcal{L}_{\text{cos}}(\boldsymbol{x}) = 1 - \frac{1}{l} \sum_{i=1}^{l} \frac{\nabla_{\boldsymbol{\theta}_i} f(\boldsymbol{x}^*, y^*) \cdot \nabla_{\boldsymbol{\theta}_i} f(\boldsymbol{x}, y)}{||\nabla_{\boldsymbol{\theta}_i} f(\boldsymbol{x}^*, y^*)||_2 ||\nabla_{\boldsymbol{\theta}_i} f(\boldsymbol{x}, y)||_2}.$$

Naturally, LAMP can also be instantiated using any other loss. Interestingly, we find that there is no loss that is universally better, and the effectiveness is dataset dependent. Intuitively, $L_1$ loss is less sensitive to outliers, while cosine loss is independent of the gradient norm, so it works well for small gradients. Thus, we set the gradient loss $\mathcal{L}_{\text{grad}}$ to either $\mathcal{L}_{\text{tag}}$ or $\mathcal{L}_{\text{cos}}$, depending on the setting.

**Embedding regularization**    In the process of optimizing the reconstruction loss, we observe the resulting embedding vectors $\boldsymbol{x}_i$ often steadily grow in length. We believe this behavior is due to the self-attention layers in transformer networks that rely predominantly on dot product operations. As a result, the optimization process focuses on optimizing the direction of individual embeddings $\boldsymbol{x}_i$, disregarding their length. To address this, we propose an embedding length regularization term:

$$\mathcal{L}_{\text{reg}}(\boldsymbol{x}) = \left( \frac{1}{n} \sum_{i=1}^{n} ||\boldsymbol{x}_i||_2 - \frac{1}{V} \sum_{j=1}^{V} ||\boldsymbol{e}_j||_2 \right)^2.$$

The regularizer forces the mean length of the embeddings of the reconstructed sequence to be close to the mean length of the embeddings in the vocabulary. The final gradient reconstruction error optimized in LAMP is given by:

$$\mathcal{L}_{\text{rec}}(\boldsymbol{x}) = \mathcal{L}_{\text{grad}}(\boldsymbol{x}) + \alpha_{\text{reg}} \mathcal{L}_{\text{reg}}(\boldsymbol{x}),$$

where $\alpha_{\text{reg}}$ is a regularization weighting factor.

**Optimization**    We summarize how described components work together in the setting of continuous optimization. To reconstruct the token sequence $\boldsymbol{t}^*$, we first randomly initialize a sequence of dummy token embeddings $\boldsymbol{x} = \boldsymbol{x}_0 \boldsymbol{x}_1 \ldots \boldsymbol{x}_n$, with $\boldsymbol{x}_i \in \mathbb{R}^d$. Following prior work on text reconstruction from gradients [3, 43], we apply gradient descent on $\boldsymbol{x}$ to minimize the reconstruction loss $\mathcal{L}_{\text{rec}}(\boldsymbol{x})$. To this end, a second-order derivative needs to be computed, as $\mathcal{L}_{\text{rec}}(\boldsymbol{x})$ depends on the network gradient at $\boldsymbol{x}$. Similar to prior work [3, 43], we achieve this using automatic differentiation in Pytorch [27].

### 4.3    Discrete Optimization

Next, we describe the discrete part of our optimization (green in Fig. 1). While continuous optimization can often successfully recover token embeddings close to the original, they can be in the wrong order, depending on how much positional embeddings influence the output. For example, reconstructions corresponding to sentences "weather is nice." and "nice weather is." might result in a similar reconstruction loss, though the first reconstruction has a higher likelihood of being natural text. To address this issue, we perform several discrete sequence transformations, and choose the one with both a low reconstruction loss and a low perplexity under the auxiliary language model.

**Generating candidates**    Given the current reconstruction $\boldsymbol{x} = \boldsymbol{x}_1 \boldsymbol{x}_2 ... \boldsymbol{x}_n$, we generate candidates for the new reconstruction $\boldsymbol{x}'$ using one of the following transformations:

- *Swap*: We select two positions $i$ and $j$ in the sequence uniformly at random, and swap the tokens $\boldsymbol{x}_i$ and $\boldsymbol{x}_j$ at these two positions to obtain a new candidate sequence $\boldsymbol{x}' = \boldsymbol{x}_1 \boldsymbol{x}_2 \ldots \boldsymbol{x}_{i-1} \boldsymbol{x}_j \boldsymbol{x}_{i+1} \ldots \boldsymbol{x}_{j-1} \boldsymbol{x}_i \boldsymbol{x}_{j+1} \ldots \boldsymbol{x}_n$.

- *MoveToken*: Similarly, we select two positions $i$ and $j$ in the sequence uniformly at random, and move the token $\boldsymbol{x}_i$ after the position $j$ in the sequence, thus obtaining $\boldsymbol{x}' = \boldsymbol{x}_1\boldsymbol{x}_2\ldots\boldsymbol{x}_{i-1}\boldsymbol{x}_{i+1}\ldots\boldsymbol{x}_{j-1}\boldsymbol{x}_j\boldsymbol{x}_i\boldsymbol{x}_{j+1}\ldots\boldsymbol{x}_n$.

- *MoveSubseq*: We select three positions $i$, $j$ and $p$ (where $i < j$) uniformly at random, and move the subsequence of tokens between $i$ and $j$ after position $p$. The new sequence is thus $\boldsymbol{x}' = \boldsymbol{x}_1\boldsymbol{x}_2\ldots\boldsymbol{x}_{i-1}\boldsymbol{x}_{j+1}\ldots\boldsymbol{x}_p\boldsymbol{x}_i\ldots\boldsymbol{x}_j\boldsymbol{x}_{p+1}\ldots\boldsymbol{x}_n$.

- *MovePrefix*: We select a position $i$ uniformly at random, and move the prefix of the sequence ending at position $i$ to the end of the sequence. The modified sequence then is $\boldsymbol{x}' = \boldsymbol{x}_{i+1}\ldots\boldsymbol{x}_n\boldsymbol{x}_1\boldsymbol{x}_2\ldots\boldsymbol{x}_i$.

Next, we use a language model to check if generated candidates improve over the current sequence.

**Using a language model to select candidates**   We accept the new reconstruction $\boldsymbol{x}'$ if it improves the combination of the reconstruction loss and perplexity:

$$\mathcal{L}_{\text{rec}}(\boldsymbol{x}') + \alpha_{\text{lm}}\mathcal{L}_{\text{lm}}(\boldsymbol{t}') < \mathcal{L}_{\text{rec}}(\boldsymbol{x}) + \alpha_{\text{lm}}\mathcal{L}_{\text{lm}}(\boldsymbol{t})$$

Here $\boldsymbol{t}$ and $\boldsymbol{t}'$ denote sequences of tokens obtained by mapping each embedding of $\boldsymbol{x}$ and $\boldsymbol{x}'$ to the nearest neighbor in the vocabulary according to the cosine distance. The term $\mathcal{L}_{\text{rec}}$ is the reconstruction loss introduced in Sec. 4.2, while $\mathcal{L}_{\text{lm}}$ denotes the perplexity measured by an auxiliary language model, such as GPT-2. The parameter $\alpha_{\text{lm}}$ determines the trade-off between $\mathcal{L}_{\text{rec}}$ and $\mathcal{L}_{\text{lm}}$: if it is too low then the attack will not utilize the language model, and if it is too high then the attack will disregard the reconstruction loss and only focus on the perplexity. Going back to our example, assume that our reconstruction equals the second sequence "nice weather is.". Then, at some point, we might use the *MoveToken* transformation to move the word "nice" behind the word "is" which would presumably keep the reconstruction loss similar, but drastically improve perplexity.

**Algorithm 1** Extracting text with LAMP

1: $\boldsymbol{x}^{(k)} \sim \mathcal{N}(0, \boldsymbol{I})$, where $k = 1, ..., n_{\text{init}}$
2: $\boldsymbol{x} \leftarrow \arg\min_{\boldsymbol{x}^{(k)}} \mathcal{L}_{\text{rec}}(\boldsymbol{x}^{(k)})$
3: **for** $i = 1$ **to** $it$ **do**
4:     **for** $j = 1$ **to** $n_c$ **do**
5:       $\boldsymbol{x} \leftarrow \boldsymbol{x} - \lambda\nabla_{\boldsymbol{x}}\mathcal{L}_{\text{rec}}(\boldsymbol{x})$
6:     **end for**
7:     $\boldsymbol{x}_{\text{best}} \leftarrow \boldsymbol{x}$
8:     $\boldsymbol{t}_{\text{best}} \leftarrow \text{PROJECTTOVOCAB}(\boldsymbol{x}_{\text{best}})$
9:     $L_{\text{best}} \leftarrow \mathcal{L}_{\text{rec}}(\boldsymbol{x}_{\text{best}}) + \alpha_{\text{lm}}\mathcal{L}_{\text{lm}}(\boldsymbol{t}_{\text{best}})$
10:     **for** $j = 1$ **to** $n_d$ **do**
11:       $\boldsymbol{x}' \leftarrow \text{TRANSFORM}(\boldsymbol{x})$
12:       $\boldsymbol{t}' \leftarrow \text{PROJECTTOVOCAB}(\boldsymbol{x}')$
13:       $L' \leftarrow \mathcal{L}_{\text{rec}}(\boldsymbol{x}') + \alpha_{\text{lm}}\mathcal{L}_{\text{lm}}(\boldsymbol{t}')$
14:       **if** $L' < L_{\text{best}}$ **then**
15:         $\boldsymbol{x}_{\text{best}}, \boldsymbol{t}_{\text{best}}, L_{\text{best}} \leftarrow \boldsymbol{x}', \boldsymbol{t}', L'$
16:       **end if**
17:     **end for**
18:     $\boldsymbol{x} \leftarrow \boldsymbol{x}_{\text{best}}$
19: **end for**
20: **return** $\text{PROJECTTOVOCAB}(\boldsymbol{x})$

### 4.4   Complete Reconstruction Attack

We present our end-to-end attack in Algorithm 1. We initialize the reconstruction $\boldsymbol{x}$ by sampling from a Gaussian distribution $n_{\text{init}}$ times, and choose the sample with minimal reconstruction loss as our initial reconstruction. Then, at each step we alternate between continuous and discrete optimization. We first perform $n_c$ steps of continuous optimization to minimize the reconstruction loss (Lines 4-6, see Sec. 4.2). Then, we perform $n_d$ steps of discrete optimization to minimize the combination of reconstruction loss and perplexity (Lines 10-17, see Sec. 4.3). Finally, in Line 20 we project the continuous embeddings $\boldsymbol{x}$ to respective nearest tokens, according to cosine similarity.

## 5   Experimental Evaluation

We now discuss our experimental results, demonstrating the effectiveness of LAMP compared to prior work in a wide range of settings. We present reconstruction results on several datasets, architectures, and batch sizes, together with the additional ablation study and evaluation of different defenses and training methods.

**Datasets**   Prior work [3] has demonstrated that text length is a key factor for the success of reconstruction from gradients. To this end, in our experiments we consider three binary classification datasets of increasing complexity: CoLA [37] and SST-2 [33] from GLUE [36] with typical sequence

lengths between 5 and 9 words, and 3 and 13 words, respectively, and RottenTomatoes [26] with typical sequence lengths between 14 and 27 words. The CoLA dataset contains English sentences from language books annotated with binary labels describing if the sentences are grammatically correct, while SST-2 and RottenTomatoes contain movie reviews annotated with a binary sentiment. For all experiments, we evaluate the methods on 100 random sequences from the respective training sets. We remark that attacking in binary classification setting is a more difficult task than in the masking setting considered by prior work [43], where the attacker can utilize strictly more information.

**Models** Our experiments are performed on different target models based on the BERT [4] architecture. The main model we consider is $BERT_{BASE}$, which has 12 layers, 768 hidden units, 3072 feed-forward filter size, and 12 attention heads. To illustrate the generality of our approach with respect to model size, we additionally consider a larger model $BERT_{LARGE}$, which has 24 layers, 1024 hidden units, 4096 feed-forward filter size, and 16 attention heads as well as a smaller model $TinyBERT_6$ from Jiao et al. [15] with 6 layers, 768 hidden units, feed-forward filter size of 3072 and 12 attention heads. All models were taken from Hugging Face [39]. The $BERT_{BASE}$ and $BERT_{LARGE}$ were pretrained on Wikipedia [5] and BookCorpus [44] datasets, while $TinyBERT_6$ was distilled from $BERT_{BASE}$. We perform our main experiments on pretrained models, as this is the most common setting for training classification models from text [25]. For the auxiliary language model we use the pretrained GPT-2 provided by Guo et al. [10], trained on the same tokenizer used to pretrain our target BERT models.

**Metrics** Following TAG [3], we measure the success of our methods based on the ROUGE family of metrics [20]. In particular, we report the aggregated F-scores on ROUGE-1, ROUGE-2 and ROUGE-L, which measure the recovered unigrams, recovered bigrams and the ratio of the length of the longest matching subsequence to the length of whole sequence. When evaluating batch sizes greater than 1, we exclude the padding tokens, used to pad shorter sequences, from the reconstruction and the ROUGE metric computation.

**Experimental setup** In all settings we consider, we compare our method with baselines DLG [43] and TAG [3] discussed in Sec. 2. As TAG does not have public code, we use our own implementation, and remark that the results obtained using our implementation are similar or better than those reported in Deng et al. [3]. We consider two variants of our attack, $LAMP_{Cos}$ and $LAMP_{L_2+L_1}$, that use the $\mathcal{L}_{cos}$ and $\mathcal{L}_{tag}$ gradient matching losses for the continuous optimization. For the $BERT_{BASE}$ and $TinyBERT_6$ experiments, we run our attack with $it = 30$, $n_c = 75$ and $n_d = 200$, and stop the optimization early once we reach a total of 2000 continuous optimization steps. For the $BERT_{LARGE}$ model, whose number of parameters make the optimization harder, we use $it = 25$ and $n_c = 200$ instead, resulting in 5000 continuous optimization steps. We run DLG and TAG for 10 000 optimization steps on $BERT_{LARGE}$ and 2500 on all other models. For the continuous optimization, we use Adam [17] with a learning rate decay factor $\gamma$ applied every 50 steps for all methods and experiments, except for $BERT_{LARGE}$ ones where, following Geiping et al. [8], we use AdamW [21] and linear learning rate decay schedule applied every step. We picked the hyperparameters for TAG, $LAMP_{Cos}$ and $LAMP_{L_2+L_1}$, separately on CoLA and RottenTomatoes using grid search on $BERT_{BASE}$ and applied them to all networks. As the optimal hyperparameters for RottenTomatoes exactly matched the ones on CoLA, we used the same hyperparameters on SST-2, as well. To account for the different optimizer used for $BERT_{LARGE}$ models, we further tuned the learning rate $\lambda$ for $BERT_{LARGE}$ experiments separately, keeping the other hyperparameters fixed. Additionally, for our methods we applied a two-step initialization procedure. We first initialized the embedding vectors with 500 random samples from a standard Gaussian distribution and picked the best one according to $\mathcal{L}_{grad}(x)$. We then computed 500 permutations on the best initialization and chose the best one in the same way. The effect of this procedure is investigated in App. C.3. Further details on our experimental setup are shown in App. D.

**Main experiments** We evaluate the two variants of LAMP against DLG [43] and TAG [3] on $BERT_{BASE}$, $BERT_{LARGE}$, and $TinyBERT_6$. Additionally, we evaluate attacks after $BERT_{BASE}$ has already been fine-tuned for 2 epochs on each task (following Devlin et al. [4]), as Balunović et al. [2] showed that in the vision domain it is significantly more difficult to attack already trained networks. For both baselines and our attacks, for simplicity we assume the lengths of sequences are known, as otherwise an adversary can simply run the attack for all possible lengths. In the first experiment we consider setting where batch size is equal to 1. The results are shown in Table 1. From the ROUGE-1

Table 1: Main results of text reconstruction from gradients with LAMP, for various datasets and architectures in the setting with batch size equal to 1. FT denotes a fine-tuned model. R-1, R-2, and R-L, denote ROUGE-1, ROUGE-2 and ROUGE-L scores respectively.

| | | BERT$_{BASE}$ | | | BERT$_{BASE}$-FT | | | TinyBERT$_6$ | | | BERT$_{LARGE}$ | | |
|---|---|---|---|---|---|---|---|---|---|---|---|---|---|
| | | R-1 | R-2 | R-L | R-1 | R-2 | R-L | R-1 | R-2 | R-L | R-1 | R-2 | R-L |
| CoLA | DLG | 59.3 | 7.7 | 46.2 | 36.2 | 2.0 | 30.4 | 37.7 | 3.0 | 33.7 | 82.7 | 10.5 | 55.8 |
| | TAG | 78.9 | 10.2 | 53.3 | 40.2 | 3.1 | 32.3 | 43.9 | 3.8 | 37.4 | 82.9 | 14.6 | 55.5 |
| | LAMP$_{Cos}$ | **89.6** | **51.9** | **76.2** | **85.8** | **46.2** | **73.1** | 93.9 | **59.3** | **80.2** | **92.0** | **56.0** | **78.8** |
| | LAMP$_{L_2+L_1}$ | 87.5 | 47.5 | 73.2 | 40.3 | 9.3 | 35.2 | **94.5** | 52.1 | 76.0 | 91.2 | 47.8 | 75.4 |
| SST-2 | DLG | 65.4 | 17.7 | 54.2 | 36.0 | 2.7 | 33.9 | 42.0 | 5.4 | 39.6 | 78.4 | 18.1 | 59.0 |
| | TAG | 75.6 | 18.9 | 57.4 | 40.0 | 5.7 | 36.6 | 43.5 | 9.4 | 40.9 | 80.8 | 16.8 | 59.1 |
| | LAMP$_{Cos}$ | **88.8** | 56.9 | **77.7** | **87.6** | **54.1** | **76.1** | **91.6** | **58.2** | **79.7** | 88.5 | **55.9** | **76.5** |
| | LAMP$_{L_2+L_1}$ | 88.6 | **57.4** | 75.7 | 41.6 | 10.9 | 39.3 | 89.7 | 53.2 | 75.4 | **89.3** | 55.5 | 75.9 |
| Rotten Tomatoes | DLG | 38.6 | 1.4 | 26.0 | 20.1 | 0.4 | 15.2 | 20.4 | 1.1 | 17.7 | 66.8 | 3.1 | 35.4 |
| | TAG | 60.3 | 3.5 | 33.6 | 26.7 | 0.9 | 18.2 | 25.8 | 1.5 | 20.2 | 73.6 | 4.4 | 36.8 |
| | LAMP$_{Cos}$ | **64.7** | **16.3** | **43.1** | **63.4** | **13.8** | **42.6** | **76.0** | **28.6** | **55.8** | 73.4 | 15.7 | 45.4 |
| | LAMP$_{L_2+L_1}$ | 51.4 | 10.2 | 34.3 | 17.2 | 1.0 | 14.7 | 74.0 | 19.4 | 46.7 | **77.6** | **16.6** | **45.8** |

metric, we can observe that we recover more tokens than the baselines in all settings. Moreover, the main advantage of LAMP is that the order of tokens in the reconstructed sequences matches the order in target sequences much more closely, as evidenced by the large increase in ROUGE-2 ($5\times$ on CoLA). This observation is further backed by the ROUGE-L metric that shows we are on average able to reconstruct up to $23\%$ longer subsequences on the BERT$_{BASE}$ model compared to the baselines. These results confirm our intuition that guiding the search with GPT-2 allows us to reconstruct sequences that are a much closer match to the original sequences. We point out that Table 1 reaffirms the observations first made in Deng et al. [3], that DLG is consistently worse in all metrics compared to both TAG and LAMP, and that the significantly longer sequences in RottenTomatoes still pose challenges to good reconstruction.

Our results show that smaller and fine-tuned models also leak significant amount of client information. In particular, TinyBERT$_6$ is even more vulnerable than BERT$_{BASE}$ and BERT$_{BASE}$-FT is shown to be only slightly worse in reconstruction compared to BERT$_{BASE}$, which is surprising given the prior image domain results. This shows that smaller models can not resolve the privacy issue, despite previous suggestions in Deng et al. [3]. Additionally, our BERT$_{LARGE}$ experiments reaffirm the observation in Deng et al. [3] that the model is highly vulnerable to all attacks.

Further, we examine the variability of our LAMP$_{Cos}$ method with respect to random initialization. To this end, we ran the BERT$_{BASE}$ experiment on CoLA with 10 random seeds, which produced R-1, R-2 and R-L of $88.2 \pm 1.02, 50.0 \pm 2.37, 75.0 \pm 1.21$, respectively, which suggests that our results are consistent. Further, we assess the variability with respect to sentence choice in App. C.1.

**Larger batch sizes** Unlike prior work, we also evaluate the different attacks on updates computed on batch sizes greater than 1 on the BERT$_{BASE}$ model to investigate whether we can reconstruct some sequences in this more challenging setting. The results are shown in Table 2. Similarly to the results in Table 1, we observe that we obtain better results than the baselines on all ROUGE metrics in all experiments, except on RottenTomatoes with batch size 2, where TAG obtains slightly better ROGUE-1. Our experiments show that for larger batch sizes we can also reconstruct significant portions of text (see experiments on CoLA and SST-2). To the best of our knowledge, we are the first to show this, suggesting that gradient leakage can be a realistic security threat in practice. Comparing the results for LAMP$_{L_2+L_1}$ and LAMP$_{Cos}$, we observe that $\mathcal{L}_{cos}$ is better than $\mathcal{L}_{tag}$ in almost all metrics on batch size 1, across models, but the trend reverses as batch size is increased.

**Sample reconstructions** We show sample sequence reconstructions from both LAMP and the TAG baseline on CoLA with $B = 1$ in Table 3, marking the correctly reconstructed bigrams with green and correct unigrams with yellow. We can observe that our reconstruction is more coherent, and that it qualitatively outperforms the baseline. In App. B, we show the convergence rate of our method compared to the baselines on an example sequence, suggesting that LAMP can often converges faster.

Table 2: Text reconstruction from gradients for different batch sizes $B$ on the $\text{BERT}_{\text{BASE}}$ model. R-1, R-2, and R-L, denote ROUGE-1, ROUGE-2 and ROUGE-L scores respectively.

| | | B=1 | | | B=2 | | | B=4 | | |
|---|---|---|---|---|---|---|---|---|---|---|
| | | R-1 | R-2 | R-L | R-1 | R-2 | R-L | R-1 | R-2 | R-L |
| CoLA | DLG | 59.3 | 7.7 | 46.2 | 49.7 | 5.7 | 41.0 | 37.6 | 1.7 | 34.0 |
| | TAG | 78.9 | 10.2 | 53.3 | 68.8 | 7.6 | 49.0 | 56.2 | 6.7 | 44.0 |
| | $\text{LAMP}_{\text{Cos}}$ | **89.6** | **51.9** | **76.2** | 74.4 | 29.5 | 61.9 | 55.2 | 14.5 | 48.0 |
| | $\text{LAMP}_{L_2+L_1}$ | 87.5 | 47.5 | 73.2 | **78.0** | **31.4** | **63.7** | **66.2** | **21.8** | **55.2** |
| SST-2 | DLG | 65.4 | 17.7 | 54.2 | 57.7 | 11.7 | 48.2 | 43.1 | 6.8 | 39.4 |
| | TAG | 75.6 | 18.9 | 57.4 | 71.8 | 16.1 | 54.4 | 61.0 | 12.3 | 48.4 |
| | $\text{LAMP}_{\text{Cos}}$ | **88.8** | 56.9 | **77.7** | 72.2 | 37.0 | 63.6 | 57.9 | 23.4 | 52.3 |
| | $\text{LAMP}_{L_2+L_1}$ | 88.6 | **57.4** | 75.7 | **82.5** | **45.8** | **70.8** | **69.5** | **32.5** | **59.9** |
| Rotten Tomatoes | DLG | 38.6 | 1.4 | 26.0 | 29.2 | 1.1 | 23.0 | 21.2 | 0.5 | 18.6 |
| | TAG | 60.3 | 3.5 | 33.6 | **47.4** | 2.7 | 29.5 | 32.3 | 1.4 | 23.5 |
| | $\text{LAMP}_{\text{Cos}}$ | **64.7** | **16.3** | **43.1** | 37.4 | 5.6 | 29.0 | 25.7 | 1.8 | 22.1 |
| | $\text{LAMP}_{L_2+L_1}$ | 51.4 | 10.2 | 34.3 | 46.3 | **7.6** | **32.7** | **35.1** | **4.2** | **27.2** |

Table 3: The result of text reconstruction on several examples from the dataset (for $\text{BERT}_{\text{BASE}}$ with $B$=1). We show only TAG (better baseline) and $\text{LAMP}_{\text{Cos}}$ as it is superior in these cases.

| | | Sequence |
|---|---|---|
| CoLA | Reference | mary has never kissed a man who is taller than john. |
| | TAG | man seem taller than mary ,. kissed has john mph never |
| | $\text{LAMP}_{\text{Cos}}$ | mary has never kissed a man who is taller than john. |
| SST-2 | Reference | i also believe that resident evil is not it. |
| | TAG | resident . or. is pack down believe i evil |
| | $\text{LAMP}_{\text{Cos}}$ | i also believe that resident resident evil not it . |
| Rotten Tomatoes | Reference | a well - made and often lovely depiction of the mysteries of friendship. |
| | TAG | - the friendship taken and lovely a made often depiction of well mysteries . |
| | $\text{LAMP}_{\text{Cos}}$ | a well often made - and lovely depiction mysteries of mysteries of friendship . |

**Ablation studies**   In the next experiment, we perform ablation studies to examine the influence of each proposed component of our method. We compare the following variants of LAMP: (i) with cosine loss, (ii) with $L_1 + L_2$ loss, (iii) with $L_2$ loss, (iv) without the language model ($\alpha_{\text{lm}} = 0$), (v) without embedding regularization ($\alpha_{\text{reg}} = 0$), (vi) without alternating of the discrete and continuous optimization steps—executing $it \cdot n_c$ continuous optimization steps first, followed by $it$ discrete optimizations with $n_d$ steps each, (vii) without discrete transformations ($n_d = 0$). For this experiment, we use the CoLA dataset and $\text{BERT}_{\text{BASE}}$ with $B = 1$. We show the results in Table 4. We observe that LAMP achieves good results with both losses, though cosine is generally better for batch size 1. More importantly, dropping any of the proposed features makes ROUGE-1 and ROUGE-2 significantly worse. We note the most significant drop in ROUGE-2 reconstruction quality happens when using transformations without using the language model ($\text{LAMP}_{\alpha_{\text{lm}}=0}$), which performs even worse than doing no transformations ($\text{LAMP}_{\text{NoDiscrete}}$) at all. This suggests that the use of the language model is crucial to obtaining good results. Further, we observe that our proposed scheme for alternating the continuous and discrete optimization steps is important, as doing the discrete optimization at the end ($\text{LAMP}_{\text{DiscreteAtEnd}}$) for the same number of steps results in reconstructions only marginally better (in ROUGE-2) compared to the reconstructions obtained without any discrete optimization ($\text{LAMP}_{\text{NoDiscrete}}$). The experiments also confirm usefulness of other features such as embedding regularization.

**Attacking defended networks**   So far, all experiments assumed that clients have not defended against data leakage. Following work on vision attacks [43, 38], we now consider the defense of adding Gaussian noise to gradients (with additional clipping this would correspond to DP-SGD [1]). Note that, as usual, there is a trade-off between privacy and accuracy: adding more noise will lead

Table 4: An ablation study with the BERT$_{BASE}$ ($B$=1) model. We restate the results for LAMP$_{Cos}$ and LAMP$_{L_2+L_1}$ from Table 1 and introduce four ablations, done on the better of the two variants of LAMP, in these cases LAMP$_{Cos}$.

| | CoLA | | | SST-2 | | | Rotten Tomatoes | | |
|---|---|---|---|---|---|---|---|---|---|
| | R-1 | R-2 | R-L | R-1 | R-2 | R-L | R-1 | R-2 | R-L |
| LAMP$_{Cos}$ | **89.6** | **51.9** | **76.2** | **88.8** | 56.9 | **77.7** | 64.7 | **16.3** | **43.1** |
| LAMP$_{L_2+L_1}$ | 87.5 | 47.5 | 73.2 | 88.6 | **57.4** | 75.7 | 51.4 | 10.2 | 34.3 |
| LAMP$_{L_2}$ | 69.4 | 30.1 | 58.8 | 72.4 | 44.1 | 65.4 | 31.9 | 5.5 | 25.7 |
| LAMP$_{\alpha_{lm}=0}$ | 86.7 | 26.6 | 66.9 | 82.6 | 37.0 | 68.4 | 64.0 | 9.9 | 40.3 |
| LAMP$_{\alpha_{reg}=0}$ | 84.5 | 38.0 | 69.1 | 83.3 | 44.7 | 71.9 | 57.8 | 11.1 | 38.3 |
| LAMP$_{DiscreteAtEnd}$ | 87.4 | 28.6 | 66.9 | 85.4 | 42.4 | 71.0 | **65.0** | 11.4 | 42.3 |
| LAMP$_{NoDiscrete}$ | 86.6 | 29.6 | 67.4 | 84.1 | 40.0 | 70.0 | 61.5 | 10.2 | 40.8 |

Table 5: Evaluation on gradients defended with Gaussian noise, with BERT$_{BASE}$ ($B$=1) on the CoLA dataset.

| | $\sigma = 0.001$ MCC= 0.551 | | | $\sigma = 0.002$ MCC= 0.526 | | | $\sigma = 0.005$ MCC= 0.464 | | | $\sigma = 0.01$ MCC= 0.364 | | |
|---|---|---|---|---|---|---|---|---|---|---|---|---|
| | R-1 | R-2 | R-L | R-1 | R-2 | R-L | R-1 | R-2 | R-L | R-1 | R-2 | R-L |
| DLG | 60.0 | 7.2 | 46.3 | 61.3 | 7.5 | 47.0 | 58.8 | 8.0 | 46.4 | 56.4 | 6.3 | 44.8 |
| TAG | 70.7 | 6.0 | 50.8 | 67.1 | 8.4 | 49.9 | 64.1 | 6.5 | 47.6 | 59.6 | 6.5 | 46.2 |
| LAMP$_{Cos}$ | **81.2** | **42.7** | **69.4** | 70.6 | 29.5 | 60.9 | 43.3 | 9.45 | 39.7 | 27.7 | 2.0 | 27.6 |
| LAMP$_{L_2+L_1}$ | 79.2 | 32.8 | 64.1 | **74.3** | **31.0** | **61.9** | **73.5** | **29.7** | **60.9** | **69.6** | **29.4** | **60.6** |

to better privacy, but make accuracy worse. We measure the performance of the fine-tuned models on CoLA using the MCC metric [23] for which higher values are better. The fine-tuning was done for 2 epochs with different Gaussian noise levels $\sigma$, and we obtained the MCC scores depicted in Table 5. We did not explore noises $> 0.01$ due to the significant drop in MCC from $0.557$ for the undefended model to $0.364$. The results of our experiments on these defended networks are presented in Table 5. While all methods' reconstruction metrics degrade, as expected, we see that most text is still recoverable for the chosen noise levels. Moreover, our method still outperforms the baselines, and thus shows the importance of evaluating defenses with strong reconstruction attacks. In App. C.2 we show that LAMP is also useful against a defense which masks some percentage of gradients.

**Attacking FedAvg** So far, we have only considered attacking the FedSGD algorithm. In this experiment, we apply our attack on the commonly used FedAvg [19] algorithm. As NLP models are often fine-tuned using small learning rates (2e-5 to 5e-5 in the original BERT paper), we find that FedAvg reconstruction performance is close to FedSGD performance with batch size multiplied by the number of FedAvg steps. We experimented with attacking FedAvg with 4 steps using $B = 1$ per step, $lr = 5e\text{-}5$ on CoLA and BERT$_{BASE}$ with LAMP$_{L_2+L_1}$. We obtained R-1, R-2 and R-L of $66.5$, $21.0$, $55.1$, respectively, comparable to the reported results on FedSGD with $B = 4$.

# 6 Conclusion

In this paper, we presented LAMP, a new method for reconstructing private text data from gradients by leveraging language model priors and alternating discrete and continuous optimization. Our extensive experimental evaluation showed that LAMP consistently outperforms prior work on datasets of varying complexity and models of different sizes. Further, we established that LAMP is able to reconstruct private data in a number of challenging settings, including bigger batch sizes, noise-defended gradients, and fine-tuned models. Our work highlights that private text data is not sufficiently protected by federated learning algorithms and that more work is needed to alleviate this issue.

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
