# OpenReview forum: "LAMP: Extracting Text from Gradients with Language Model Priors"
_NeurIPS.cc/2022/Conference — NeurIPS 2022 Accept_

### Official Review · Reviewer_24M9 · 2022-06-22

**Rating:** 6
**Confidence:** 3
**Soundness:** 3 good
**Presentation:** 2 fair
**Contribution:** 3 good

**Summary:**

This paper proposes a novel strategy to attack pretrained models for text models which aims to reconstruct the private user data used to finetune the model through federated learning. The algorithm takes the finetune gradient as well as the target label as the input, then search for the private user input sentence. The general idea is a iterative two-step process, where the first step is a continuous optimization to search for a embedding that leads to similar gradient, and the second step is a discrete optimization that uses GPT models to find a output sentence with the lowest perplexity.

####
Thanks for the authors' response! I think it might be worth it to clarify the main novelty of the paper in the main method section. It would also be nice to clearly discuss the limitation at the end of the paper, as researchers not familiar with this line of research might benefit from some background information about the social impact/limitation of the method.

**Questions:**

1. Could you clarify the major difference between your method and prior work?
2. How can this method be extended to the more realistic setting where one does not have labels for the data?

**Limitations:**

The authors did not address limitations and potential negative impact. There are several points worth mentioning:
1. Works on adversarial attack could be exploited by hackers.
2. The method is only limited to classification data.

**Strengths And Weaknesses:**

strengths:
1. the method is very interesting and novel. It also addresses an interesting problem in the pretrain-finetune paradigm that is very popular now.
2. the writing is mostly well structured and easy to understand.
3. the final performance also seems good compared to previous methods. The ablation study and examples are pretty nice addition to the results.

weakness:
1. the paper didn't really clarify the main difference between the proposed method and the prior work, such as TAG. Therefore, it is hard to tell what are the exact novelty the paper adds and the effect of the novelty. My understanding is that the random transformation and the use of GPT models allow the output more natural. However, it is really hard to tell from the current structure of the paper.
2. the method can only be applied to classification where label is known, which seems pretty limited. It might be more natural to have unlabeled data for federated learning setting, as one might not be able to annotate the user text.

---

> ### Author Response · Authors · 2022-08-02
> **Answer to Reviewer 24M9 Part 1/2**
>
> We thank the reviewer for recognizing the importance of the problem we are solving. We provide the requested discussion on limitations and contributions of our attack method below. Further, as we point out below, we have added to the supplementary material additional discussion on the main differences between LAMP and TAG.
>
> **Q:** What are the main differences between LAMP and TAG?
>
> **A:** The main differences between LAMP and TAG are:
> - The error function $\mathcal{L}\_\text{rec}$ used in the continuous part of the optimization that results in better token reconstruction (increased R-1).
>     - In particular, we introduce the regularization term $\mathcal{L}\_\text{reg}$ that helps the continuous optimization.
>     - We show that the cosine error function, previously applied in the image domain, can often outperform the error function suggested by TAG for text reconstruction.
> - The introduction of our discrete optimization that takes advantage of the GPT model to help reconstruct the token order better, resulting in a better R-2 measure. Our discrete optimization step is novel in several ways:
>     - In particular, we introduce several possible discrete transformations that help fixing common token ordering problems arising from the continuous reconstruction.
>     - We choose to alternate our discrete and continuous optimization steps, resulting in better token order reconstruction (refer to our answer to Reviewer peFq).
>     - We take advantage of the perplexity computed by the existing GPT-2 language model to evaluate the different discrete transformations.
>
> We have added a new section (Section A.2) in the latest version of the supplementary material, outlining these differences.
>
> **Q:** How can this method be extended to the more realistic setting where one does not have labels for the data?
>
> **A:** [6] considers the setting where the text model is trained for next-word prediction instead of the classification (which we consider), showing that in this setting one can use label reconstruction techniques to obtain the set of input tokens (but not their order). Despite [6] operating in an easier setting (from an attacker's point of view), where the server is allowed to send malicious weights to the clients, this part of their method can also be applied in the setting we consider, where the server is honest-but-curious.
>
> This makes the problem of reconstructing data in the next-word prediction setting simpler to solve compared to reconstruction in the classification setting. To this end, we have focused on the harder classification setting. However, our method is easy to combine with the label reconstruction from [6] to obtain a gradient leakage algorithm in the next-word prediction setting.
>
> **Q:** Can you more explicitly discuss the limitations and assumptions of LAMP, and comment on adversarial attacks and problem settings beyond classification?
>
> **A:** Yes. As outlined in Appendix A, LAMP focuses on the setting where the server is honest-but-curious, the text model is trained for classification, and labels are known. As adversarial attacks consider a different threat model to gradient leakage in federated learning, it is not directly clear how they are directly relevant to the discussion of limitations of LAMP. We would appreciate it if the reviewer could more explicitly point out the connection, and we are happy to continue the discussion on this.
>
> Further, regarding limitations of the attack, in Table 4 and Table 6 (in supplementary material), we demonstrate that clients can defend against our attacks by pruning and/or adding noise to their gradients in exchange of a lower final model accuracy; our experiments in Table 1 also suggest that another possible client defense against our attack is to use big batch sizes during training. We believe the latter is likely a more practical defense against our attack, as it will result in better final model accuracy. We will make the explanations sharper in the next revision.

---

> ### Author Response · Authors · 2022-08-02
> **Answer to Reviewer 24M9 Part 2/2**
>
> **Q:** Can you include a discussion of the negative societal impact of your work?
>
> **A:** The current version of the paper notes (in the Checklist) that LAMP operates in the same setting as referenced closely related prior work, and thus no fundamentally new societal issues are being brought up by our paper.
>
> To make this more explicit, like previous gradient leakage attacks ([3,7,36,37,39]), our attack can be used to compromise the privacy of client data in real-world federated learning setups, especially when no defenses are used by the clients. Our attack emphasizes that text data, similarly to data in other domains, is vulnerable to gradient leakage attacks and that when federated learning is applied in practice extra steps need to be taken to mitigate the potential risks. Further, in line with the related work, we study possible mitigations to our attack’s success in Tables 1, 4 and 6 in the paper, thus promoting possible practical federated learning implementations that will be less vulnerable.
>
> We welcome any further concrete suggestions on how to expand upon this discussion and we will be happy to incorporate them alongside the extended discussion given above in the next revision of the paper.

---

### Official Review · Reviewer_peFq · 2022-07-11

**Rating:** 6
**Confidence:** 3
**Soundness:** 3 good
**Presentation:** 4 excellent
**Contribution:** 3 good

**Summary:**

The authors propose a model for recovering user data from gradient updates in a federated learning system for text classification. They achieve this by alternating continuous gradient based optimization with discrete heuristic based token-reshuffling. The authors show that the proposed model outperforms methods that use only gradient based updates to the tokens.

**Questions:**

How important is to alternate token-reshuffling updates with continuous optimization? What if the token-reshuffling appears only at the end of the continuous updates?

**Limitations:**

The limitations haven’t been discussed.

**Strengths And Weaknesses:**

The primary contribution of the paper is the alternation between gradient-based updates and token reshuffling for reconstructing user text. I am unaware if such an approach has been attempted for adversarial attacks in text. Hence, this can be assumed to be novel.

The paper is well-written and easy to follow. The ablation studies confirm the importance of token-reshuffling for learning a good attack. The ablation studies also show the importance of L1+L2 loss (although this was proposed in a different paper).

From a novelty perspective, this alternation between token reshuffling and continuous updates for attacking text classifier appears to be novel. However, I am not an expert in this field.

Some parts of the paper are unclear. For instance, the authors mention that they use Adam for learning the embeddings for each input during the continuous optimization phase. However, they do not mention how they compute the gradient of the loss with respect to the input embeddings x*. The gradient computation requires second order derivatives which hasn’t been discussed in the paper at all.

It is unclear how token-reshuffling with continuous gradient-based updates influences the descent direction. Specifically, I am wondering if token-reshuffling is only performed at the end of all gradient-based updates, how will the performance be affected.

---

> ### Author Response · Authors · 2022-08-02
> **Answer to Reviewer peFq**
>
> We thank the reviewer for acknowledging the novelty of our method and the clarity of our writing. We detail below the requested experimental results and clarifications alongside requested changes made to the paper. For the discussion of limitations of our attack, refer to the answer provided to reviewer 24M9.
>
> **Q:** How are the gradients of the loss $\mathcal{L}\_\text{rec}$ with respect to the input embeddings $x$ computed? Are second order derivatives needed to compute them?
>
> **A:** Yes, a second order derivative needs to be computed for the continuous part of the optimization in our algorithm (Line 5 in Algorithm 1). Our algorithm follows previous work (see [3,7,36,37,39]) and relies on Pytorch’s automatic differentiation to compute these second order derivatives that are then used by Adam to optimize the embeddings. We have clarified this point in the latest version of our paper (Line 173).
>
> **Q:** Can you provide experiments where the token-reshuffling appears only at the end of the continuous updates?
>
> **A:** We experimented with this on the $\text{BERT}_\text{BASE}$ model on batch size of 1. We compare our original method (denoted $\text{LAMP}\_\text{Cos}$) with a version of the method where the token transformations are applied after the continuous optimization (denoted Continuous+Discrete). We use the same number of discrete optimization steps for both algorithms. The results are presented below and added to the latest revision of the supplementary material of the paper. We observe that while applying the discrete optimization at the end results in the similar number of recovered tokens, their order is much worse as demonstrated by the significant drop in the R-2 measure.
> $$
>   \begin{array}{rlllllllll}
>     & \hspace{2.5cm} \rlap{\text{CoLA}} &&& \hspace{2.5cm} \rlap{\text{SST-2}}  &&& \hspace{1.4cm} \rlap{\text{Rotten Tomatoes}} \\\\
> & \rlap{\underline{\hspace{6.0cm}}} &&&  \rlap{\underline{\hspace{6.0cm}}}  &&&  \rlap{\underline{\hspace{5.6cm}}}\\\\
>       & \qquad \text{ R-1 } & \   \text{ R-2 }  & \qquad  \text{ R-L } & \quad  \text{ R-1 } & \  \text{ R-2 }  &  \qquad  \text{ R-L } & \quad  \text{ R-1 } & \qquad  \text{ R-2 }  & \qquad  \text{ R-L } \\\\
>     \hline
> \text{LAMP}\_\text{Cos} & \hspace{0.8cm} \textbf{89.6} & \hspace{0.2cm}  \textbf{51.9} &  \hspace{0.87cm} \textbf{76.2} & \hspace{0.45cm} \textbf{88.8} &  \hspace{0.1cm}  \textbf{56.9} & \hspace{0.87cm}  \textbf{77.7} &  \hspace{0.45cm}  64.7 & \hspace{0.8cm}  \textbf{16.3} &  \hspace{0.87cm}  \textbf{43.1}\\\\
> \text{Continuous+Discrete} & \hspace{0.8cm} 87.4 & \hspace{0.2cm}  28.6 &  \hspace{0.87cm} 66.9 & \hspace{0.45cm} 85.4 &  \hspace{0.1cm} 42.4 & \hspace{0.87cm} 71.0 &  \hspace{0.45cm} \textbf{65.0} & \hspace{0.8cm} 11.4 &  \hspace{0.87cm} 42.3
>  \end{array}
> $$

---

### Official Review · Reviewer_ZC5x · 2022-07-12

**Rating:** 7
**Confidence:** 3
**Soundness:** 3 good
**Presentation:** 4 excellent
**Contribution:** 3 good

**Summary:**

This paper focuses on the attack that tries to recover text data from the gradients, this type of attack is a threat particularly for federated learning where the central server may recover the private client data through gradients. Based on previous work that optimises the input to minimise the distance between gradients, this paper further proposes to alternate continuous optimization and discrete optimization that is guided through a language model prior. The discrete optimization is claimed to help obtain text data that is more like fluent language. The resulting approach, LAMP, greatly outperforms previous approaches on three binary-classification benchmarks.


**Questions:**

I listed all my questions or suggestions in the weaknesses above, I would like to improve my rating if the authors could address them in the rebuttal, particularly the *major* points.

**Limitations:**

Yes

**Strengths And Weaknesses:**

### Strengths:

1. The proposed approach is novel and technically sound. The motivation and contribution are clear – from the examples in Table 2 and quantitative improvements, it does seem like previous approaches fail to yield grammatical text while the proposed discrete optimization step seems to help it a lot.
2. The empirical results are strong especially with batch size > 1. The ablation study is appreciated as well.
3. The paper is well-written.



### Weaknesses:

1. *(minor)* The proposed approach adds several additional hyperparameters to tune, for  example, $\alpha_{lm}, \alpha_{reg}, n_c, n_d, n_{init}$. This surely adds complexity of the approach and may make the proposed attack difficult to be practically applied. While I appreciate the detailed hyperparameter paragraph in Line 268-274, I think it would be better to report the hyperparameter selection range as well, so that the readers could have an idea how much effort is imposed to tune these hyperparameters.

2. *(major)* I am a bit worried that the model comparison is only conducted on a randomly selected set of 100 sentences. 100 sentences sounds too few for me, and I am not sure how robust the model rank is based on only 100 random examples. I feel this point should be justified properly, either through constructing a larger test set, or using different random seeds to generate different sets of 100 test examples and testing on each of them.

3. *(minor)* While Line 239 mentions to illustrate the generality of the proposed approach with respect to model size, I don’t think this paper really made that goal by only using tiny and base sizes of BERT – at least BERT-large should be included to have a relatively complete coverage. I understand that the authors may be limited by resources to use larger models, which is fine. I just wanted to point out the current experiments are not sufficient to indicate generality with respect to model sizes (this is rather a minor point anyway).

4. *(major)* Line 257-259 mentions the baselines use 2500 iterations while the proposed approach uses 2000. Is *it* in Algorithm 1 2000 (and what are the values of *n_c* and *n_d*)? How did you choose these numbers? Do the baselines fully converge? I would like to see more justifications to show that the comparison is fair, because LAMP employs a nested for loop (while the baselines do not?) in optimization and I feel the number of total optimization steps (and the cost/time) in LAMP is actually larger than the baselines, right?

5. *(minor)* Line 272 mentions that LAMP additionally adopts a two-step initialization procedure that seems important to me. I would like to see the ablation results on this two-step init in Table 3 to know how much of the improvement over baselines is from this initialization.


```
After author response:
The author response addressed most of my concerns, and I would like to increase my score to 7 given that the authors will update the paper accordingly as promised.
```

---

> ### Author Response · Authors · 2022-08-02
> **Answer to Reviewer ZC5x Part 1/2**
>
> We thank the reviewer for their insightful comments and questions. We address all concerns below. Further, we have updated our paper in order to reflect the provided comments. We explicitly point to these changes in the relevant answers below.
>
> **Q:** How many iterations do you use in your algorithm, and how were these numbers chosen? How do the runtimes of your method and the baselines compare?
>
> **A:** The reported 2000 iterations is the total number of continuous optimization steps used by our algorithms. To this end, we set $\text{it}=30$ and $n\_c=75$ and stop the optimization early once we reach a total of 2000 continuous optimization steps. We have clarified this in the latest version of our paper (L260-261). $n\_c$ was chosen such that it is big enough for the continuous optimization to be able to make significant changes to the embeddings between two consecutive discrete optimization steps, but also small enough that the discrete optimization is executed often and thus can influence the end-to-end optimization procedure.
>
> Further, our methods use $n_d=200$, as we experimentally found that 200 random transformations are enough to find near-optimal token ordering according to the error function $\mathcal{L}_\text{rec}(x’ ) + \alpha_\text{lm}\mathcal{L}_\text{lm}(t’)$ for the sentence lengths present in the our datasets. We chose the total number of continuous optimization steps for both our method (2000) and the baseline method (2500) such that the methods converge. We note that while our algorithm executes less continuous optimization steps in total compared to the baselines, our method’s runtime is around $\approx 1.5x$  more than the baselines due to the additional computational cost of the discrete optimization. We plan to provide precise runtimes of the different methods in the next revision of the paper.
>
> **Q:** How does the two-step initialization procedure affect the results?
>
> **A:** We have extended our ablation study and investigated the requested comparison between reconstruction using a single random initialization ($n_{init}=1$), and the one using our two-step initialization procedure with $n_{init}=500$, on the $\text{BERT}_\text{BASE}$ model with our $\text{LAMP}\_\text{Cos}$ method and batch size of 1. The results are shown below. We observe that the two-step initialization scheme consistently improves individual token recovery (measured in terms of R-1) but may in some cases slightly degrade token ordering results (measured in terms of R-2). Even though we used two-step initialization in the paper (it is strictly better on one dataset and non-comparable to single random initialization on the remaining datasets), it is indeed sometimes possible to get better results with the latter. We will expand our experimental results accordingly in the updated version.
>
> $$
>   \begin{array}{rlllllllll}
>     & \hspace{2.5cm} \rlap{\text{CoLA}} &&& \hspace{2.5cm} \rlap{\text{SST-2}}  &&& \hspace{1.4cm} \rlap{\text{Rotten Tomatoes}} \\\\
> & \rlap{\underline{\hspace{6.0cm}}} &&&  \rlap{\underline{\hspace{6.0cm}}}  &&&  \rlap{\underline{\hspace{5.6cm}}}\\\\
>     n_\text{init}  & \qquad \text{ R-1 } & \   \text{ R-2 }  & \qquad  \text{ R-L } & \quad  \text{ R-1 } & \  \text{ R-2 }  &  \qquad  \text{ R-L } & \quad  \text{ R-1 } & \qquad  \text{ R-2 }  & \qquad  \text{ R-L } \\\\
>     \hline
> 500 & \hspace{0.8cm} \textbf{89.6} & \hspace{0.2cm}  \textbf{51.9} &  \hspace{0.87cm} \textbf{76.2} & \hspace{0.45cm} \textbf{88.8} &  \hspace{0.1cm} 56.9 & \hspace{0.87cm} 77.7 &  \hspace{0.45cm} \textbf{64.7} & \hspace{0.8cm} 16.3 &  \hspace{0.87cm} 43.1\\\\
> 1  & \hspace{0.8cm} 87.3 & \hspace{0.2cm}  48.1 &  \hspace{0.87cm} 73.2 & \hspace{0.45cm} 87.4 &  \hspace{0.1cm} \textbf{60.8} & \hspace{0.87cm} \textbf{78.8} &  \hspace{0.45cm} 63.7 & \hspace{0.8cm} \textbf{16.6} &  \hspace{0.87cm} \textbf{43.8}
>  \end{array}
> $$

---

> > ### Comment · Reviewer_ZC5x · 2022-08-07
> > **Thanks for the response!**
> >
> > Thank you for the response! I am satisfied with the response, and I would like to increase my score to 7 given that the authors will update the paper accordingly as promised.

---

> ### Author Response · Authors · 2022-08-02
> **Answer to Reviewer ZC5x Part 2/2**
>
> **Q:** How are the values of the hyperparameters of the proposed method chosen? What ranges did you use for different hyperparameters in the grid search procedure described in Lines 268-274?
>
> **A:** We experimentally found that our algorithm is robust with respect to the exact values of $n_c$ and $n_d$, provided that the choice follows the guidelines outlined in the previous questions. Further, as demonstrated by our experiments above the performance of our algorithm changes minimally with $n_\text{init}$. To this end, we did not include $n_c$, $n_d$ and $n_\text{init}$ in the grid search procedure.
>
> For both $\text{LAMP}\_\text{Cos}$ and the baselines we investigated the ranges of $\alpha\_\text{lm} \in  [0.05, 0.2]$, $\alpha\_\text{reg} \in  [0.01, 1]$, $\lambda \in [0.001, 0.5]$, $\gamma \in  [0.8, 1]$, and $\alpha_{tag} \in [10^{-5}, 10^{2}]$. For $\text{LAMP}\_{\text{L}\_2+\text{L}\_1}$ , we consider $\alpha\_\text{lm} \in [30, 240]$ and $\alpha\_\text{reg} \in [10, 100]$, as the scale of loss values is orders of magnitude larger than in $\text{LAMP}\_\text{Cos}$. We note that compared to TAG and DLG, the only additional hyperparameters we need to grid search over are $\alpha_{reg}$ and $\alpha_{lm}$. We have reported the grid search ranges in Appendix D in the latest version of the supplementary material.
>
> **Q:** Can you confirm the experimental conclusions on test examples sampled using different random seeds?
>
> **A:** Yes. As suggested by the reviewer, we ran the $\text{BERT}_\text{BASE}$ CoLA experiment with B=1 on additional 10 different subsets of 100 sentences randomly chosen from the test sets. The mean $\pm$ one standard deviation for R-1, R-2 and R-L are given in the table below. We see the results are mostly consistent between subsets.
> $$
>   \begin{array}{rccc}
>     & \text{ R-1 } & \text{ R-2 }  & \text{ R-L } \\\\
>     \hline
> \text{DLG}	& 56.2\pm5.0 &	6.5\pm1.6 & 45.0\pm2.6 \\\\
> \text{TAG}	& 74.4\pm3.1 &	10.7\pm1.8 & 53.0\pm2.1 \\\\
> \text{LAMP}\_\text{Cos}	& 87.8\pm2.6 &	48.4\pm5.5  & 74.6\pm2.9 \\\\
> \text{LAMP}\_{\text{L}_2+\text{L}_1}	& 83.1\pm3.7 &	40.7\pm5.7 & 69.3\pm3.6 \\\\
>  \end{array}
> $$
>
> **Q:**  Can you include experiments on bigger models, e.g. $BERT_\text{Large}$?
>
> **A:** We appreciate the suggestion, and agree that experiments with $BERT_\text{Large}$ would further strengthen our claim that LAMP is general with respect to model size. However, obtaining reliable and comprehensive results is out of scope for this rebuttal given our resources, as ensuring adequate parameter choices for each method is computationally intensive (10s-100s of runs with ~1.5 day latency).
>
> Importantly, note that prior work consistently points out that models of all sizes are vulnerable to gradient inversion attacks [7, 36], with commonly minor differences in attack strength across sizes. Thus, we expect the results on $BERT_\text{Large}$ to be consistent with our current findings.

---

### Meta-Review · Area_Chair_KRfX · 2022-08-23

**Recommendation:** Accept
**Confidence:** Certain

**Metareview:**

This paper describes a novel method to recover the input text based on the computed gradient. This is important in the context of federated learning, which promises to enable learning through gradient sharing while keeping the input text secret. The findings of the paper demonstrate that gradients are sufficient to recover significant parts of the input text questioning the federated learning premise at least in the context of large language models.

The approach in novel and technically sound. Empirical results are convincing. The paper is well-written and clear.

Given current trends to growing model size, it will be great if the paper can further scale the experimental results to larger models.

**Award:**

No

---

### Decision · Program_Chairs · 2022-09-14

Accept